# Innate and Adaptive Immune Responses in the Upper Respiratory Tract and the Infectivity of SARS-CoV-2

**DOI:** 10.3390/v14050933

**Published:** 2022-04-29

**Authors:** Ranjan Ramasamy

**Affiliations:** ID-FISH Technology, 556 Gibraltar Drive, Milpitas, CA 95035, USA; rramasamy@idfishtechnology.com

**Keywords:** adaptive immunity, age and SARS-CoV-2 infectivity, COVID-19 vaccines, human genetics and SARS-CoV-2 infectivity, innate immunity, intranasal immunization, nasal epithelium, upper respiratory tract immunity

## Abstract

Increasing evidence shows the nasal epithelium to be the initial site of SARS-CoV-2 infection, and that early and effective immune responses in the upper respiratory tract (URT) limit and eliminate the infection in the URT, thereby preventing infection of the lower respiratory tract and the development of severe COVID-19. SARS-CoV-2 interferes with innate immunity signaling and evolves mutants that can reduce antibody-mediated immunity in the URT. Recent genetic and immunological advances in understanding innate immunity to SARS-CoV-2 in the URT, and the ability of prior infections as well as currently available injectable and potential intranasal COVID-19 vaccines to generate anamnestic adaptive immunity in the URT, are reviewed. It is suggested that the more detailed investigation of URT immune responses to all types of COVID-19 vaccines, and the development of safe and effective COVID-19 vaccines for intranasal administration, are important needs.

## 1. Background

Coronavirus disease 2019 (COVID-19) is caused by infection with the severe acute respiratory syndrome coronavirus 2 (SARS-CoV-2). As of 20 April 2022, there have been 506 million COVID-19 cases with 6 million deaths [1], and severe economic and social disruption worldwide, after the first identification of COVID-19 in December 2019. SARS-CoV-2 is a membrane-enveloped virus with a 30 kb positive-sense RNA genome and is related to two recent highly pathogenic coronaviruses of zoonotic origin (SARS-CoV-1 and MERS-CoV) and several less pathogenic coronaviruses that commonly cause cold-like symptoms [2]. The spike glycoprotein (S) in the viral envelope contains a receptor-binding domain (RBD) in its N-terminal S1 domain for binding the angiotensin-converting enzyme 2 (ACE2) receptor on respiratory epithelial cells, and a C-terminal S2 region that subsequently helps fuse the virus and host cell membranes to allow the entry of viral RNA into the cytoplasm [2,3]. Host cell proteases furin and TMPRSS2 are, respectively, important for cleaving the S protein at the S1/S2 junction to facilitate ACE2 binding and within the S2 to activate membrane fusion [4]. SARS-CoV-2 can alternatively enter cells by endocytosis followed by the S-mediated fusion of the endosome and virus membrane [4]. S also promotes membrane fusion between host cells to directly spread infection [4].

The nasal epithelium in the upper respiratory tract (URT) is the initial site of infection via inhaled SARS-CoV-2 virions [5,6,7,8,9,10,11]. Early and effective antigen non-specific, innate immune responses limit viral replication and help eliminate the virus in the URT [12,13,14,15]. Innate immune responses also orchestrate subsequent adaptive immune responses involving antibodies produced by plasma cells derived from bone-marrow-derived B lymphocytes (B cells) and effector thymus-processed lymphocytes (T cells) in local lymphoid tissue a few days after infection, which contribute to resolving the infection [14,15,16]. These processes are illustrated in Figure 1. Healthy persons can eliminate SARS-CoV-2 with either no detectable symptoms or only mild COVID-19 symptoms (defined here as not requiring hospital treatment) as a result of rapid and effective innate and adaptive immune responses in the URT [12,13,14,15,16].

Protective innate immunity in the URT typically involves: (i) the barrier function of the mucus layer; (ii) the binding of virions by mucins as well as antibacterial proteins and peptides such as defensins and collectins in the mucus; (iii) complement activation by the alternative and lectin pathways through the recognition of altered surfaces in virions and infected cells to promote lysis, opsonization and inflammation; (iv) the induction of an anti-viral state in infected and neighboring epithelial cells through the inhibition of protein synthesis and mRNA degradation, and the activation of phagocytic cells and antigen-presenting dendritic cells. This entails pathogen-associated molecular patterns (PAMPs) being recognized by cellular pattern recognition receptors (PRRs) and the production of types 1 and 3 interferons (IFNs); (v) the activation of macrophages and dendritic cells, resulting in the production of cytokines such as IL-1, IL-6, IL-12, IL-18 and TNF that promote an inflammatory response locally and systemically, activate natural killer (NK) cells to lyse virus infected cells and promote adaptive immune responses; (vi) innate T cells recognizing damage-associated molecular patterns (DAMPs) in infected epithelial cells and secreting type 2 IFNγ, which further activates NK cells, phagocytes, dendritic cells and the adaptive immune response [15].

Key effectors in antigen-specific adaptive immunity are CD4+ helper T cells (T_H_), CD8+ cytotoxic T cells (T_C_) and antibodies secreted by plasma cells. Dimeric IgA antibodies transported across the mucosal epithelium into the URT airway can bind proteins exposed on the virion membrane, notably S, agglutinate virions and prevent virions binding to epithelial cells. IgG and IgM antibodies in the nasal mucosa can also similarly neutralize virions and additionally activate complement through the classical pathway to lyse virions and infected epithelial cells. IgG antibodies can moreover promote Fc receptor-mediated phagocytosis of virions [15]. T_H_ cells in URT lymphoid tissues activate B cells and promote immunoglobulin class switching and the affinity maturation of antibodies. They also secrete cytokines such as IFNγ that activate phagocytes and NK cells, upregulate major histocompatibility complex (MHC) molecules on antigen-presenting cells and promote T_C_ function. T_H_ cells can also directly kill virus-infected cells which, however, is the major function of T_C_ cells [16]. A proportion of B and T cells responding to infection with SARS-CoV-2 or COVID-19 vaccination differentiate to memory cells that may partly reside in the URT and are responsible for a faster and more effective anamnestic immune response upon subsequent exposure to the virus [16].

The importance of adequately warming and humidifying inhaled air in the nose (termed nasal air conditioning), a process that is influenced by age, air quality, gender and host genetics, for generating optimal innate and adaptive immune responses against SARS-CoV-2 in the URT was recently highlighted [14,15]. The significance of nasal air conditioning in resisting infection is further supported by evidence that the S protein of the Wuhan strain of SARS-CoV-2 promotes better infectivity at 33 °C than 37 °C [17]. The nasal conditioning of inhaled air at 25 °C produces a temperature of approximately 33 °C in the nasal cavity, which will be further lowered when the ambient air temperature falls below 25 °C [14,15]. The effect of nasal cavity temperatures lower than 33 °C on the susceptibility of the nasal epithelium to infection with SARS-CoV-2, though not experimentally established, is predicted to adversely affect innate and adaptive immune responses in the URT [14,15].

The protective URT immune responses to infection with the human influenza A virus, which is a negative-strand RNA virus, has been studied over a longer period [18,19,20,21] than for SARS-CoV-2. Molecular signaling processes underlying innate immune responses in the URT epithelial cells to influenza A infection, outlined in Figure 2, have been a paradigm for understanding the corresponding signaling during SARS-CoV-2 infection.

The influenza A virus replicating in the nasal epithelium reaches the pharynx through mucociliary clearance [22], and then the lower respiratory tract (LRT) and lungs via the oral–lung axis [23]. The same processes are also likely to occur with SARS-CoV-2. The infection of the LRT with SARS-CoV-2 can lead to severe pneumonia, acute respiratory distress syndrome and death [24,25]. Many more factors influence SARS-CoV-2 infection of the LRT and the attendant development of severe COVID-19 with systemic co-morbidities and lung-specific pathology than infection of the URT. Controlling replication and eliminating SARS-CoV-2 at the initial sites of infection in the URT is therefore critically important for preventing severe COVID-19 and also reducing SARS-CoV-2 transmission. The effectiveness of immune responses to SARS-CoV-2 in the URT depends on whether a person has previously been infected with SARS-CoV-2 or not and whether that person has been vaccinated. Vaccination against COVID-19 may, by improving early URT immunity, ameliorate the enhanced susceptibility to SARS-CoV-2 that results from weaker nasal air conditioning of colder air in winter [15,26]. The rapid progress being made in understanding protective innate and adaptive immune responses to SARS-CoV-2 in the URT in these different contexts is reviewed in this article. Figure 3 presents an overview of the many factors that influence immunity to SARS-CoV-2 in the URT.

## 2. Importance of Immunity in the URT for Rapidly Resolving SARS-CoV-2 Infections

### 2.1. Studies on Early Innate Immune Responses in Nasal Epithelial Cell Cultures

Early innate immune defenses in the respiratory tract are important for combatting many respiratory virus infections [27]. Knowledge of the early innate and adaptive immune responses in the URT during SARS-CoV-2 infections has progressed rapidly since a recent review [15]. New studies with primary human nasal epithelial cell cultures have been particularly illuminating in this regard [28]. Single-cell RNA-seq and proteomics analyses showed the rapid production of type 1 and 3 IFNs in response to Sendai virus infection of the cultures, while the IFN responses were delayed during SARS-CoV-2 infection, so that SARS-CoV-2 gene expression reached maximum levels before the upregulation of type 1 IFN gene expression in the cultures [28]. Infection of the cultures with SARS-CoV-2, however, produced a progressive upregulation of genes for the proinflammatory cytokines IL6, TNFα and IL1β, demonstrating that the virus was being recognized within the cells [28]. Pre-treatment of the cultured nasal epithelia cells for 16 h before infection or up to 6 h post-infection with type 1 IFNs reduced SARS-CoV-2 infectivity, demonstrating the potential efficacy of type 1 IFNs in URT immunity to SARS-CoV-2, and indicating a possible use in therapy and prophylaxis [28]. These observations support the accumulating evidence that SARS-CoV-2 can interfere with protective URT innate immune responses in multiple ways (discussed further in Section 3).

### 2.2. Early Innate Immune Responses in the Nasal Mucosa of Children and Adults

Children have a lower risk of developing severe COVID-19 than adults [29,30]. This can be due to immune responses in children being better able to control and eliminate SARS-CoV-2 infection in the URT. Recent findings support this assertion. Uninfected children were reported to have resident immune cells with similar gene expression profiles to uninfected adults in the nasal mucosa except for significantly greater numbers of CD8+ cytotoxic T_C_ cells with a tissue-resident memory phenotype [31]. In contrast, another study on uninfected and infected individuals, using single-cell RNA-seq on nasal swabs, found higher numbers of many types of immune cells, notably neutrophils, in the nasal mucosa of healthy children but also lower numbers of ciliated epithelial cells, a prominent cell type for SARS-CoV-2 replication, as well as similar levels of ACE2 and cellular proteases involved in cleaving S for viral entry [32]. Importantly, children had higher basal expression levels of viral RNA PRR genes, including *RIG-1* and *MDA5*, that also increased more rapidly than in adults for up to five days post-infection with SARS-CoV-2. The increase in PRRs was accompanied by the markedly greater expression of type 1 IFN-stimulated ISGs in children during infection of the nasal mucosa. The small numbers of macrophages and dendritic cells in the nasal mucosa of children also showed the greater basal level expression of inflammatory cytokine and chemokine genes for IL1β, IL8, TNF, CCL3 and CCL4 [32]. CD8+ T_C_ cells in the nasal mucosa with a resident memory phenotype expressing higher levels of the type 2 IFNγ transcripts in children than adults before and after infection were also observed [32]. Compatible results were obtained in another recent study that investigated cellular gene expression and nasal fluid protein levels in nasal swabs from children and adults presenting with COVID-19 in US hospitals [33]. Significant differences in the expression of ACE2 and TMPRSS2 genes, SARS-CoV-2 copies or levels of mucosal IgA and IgG antibodies to SARS-CoV-2 between adults and children were not found in this study, but children showed the more pronounced expression of innate immune response genes (e.g., for type 1 IFNα) and inflammatory genes (for IFNγ, IL1β, NLRP3, IL8 and IP10/CXCL10), significantly higher levels of IFNα2, IFNγ, IP10/CXCL10 and IL1β in nasal secretions, as well as the greater expression of genes associated with adaptive immune response cells (*CD4, CD8α* and *CD20*) [33]. Adults and children with COVID-19 and healthy age-matched controls in the UK were also recently investigated using a single-cell, multi-omic approach [34]. The findings further confirmed that the healthy URT of children was already in a more type 1 IFN-activated state than healthy adults, and that the difference in type 1 IFN responses was further enhanced, notably in nasal mucosal dendritic cells, upon SARS-CoV-2 infection [34]. The T cell antigen receptor repertoire in children was also broader than in adults, reflecting a more infection-naïve phenotype, and it was suggested that this can lead to a faster and more efficient adaptive immune response in the URT in response to SARS-CoV-2 infection [34].

Serial nasopharynx sampling in patients confirmed that the early induction of many ISGs and IP10/CXCL10 in the URT is important for restricting SARS-CoV-2 replication early in an infection and preventing the development of severe COVID-19 [35]. Single-cell RNA-seq analysis of cells in nasopharyngeal swabs of patients presenting early with COVID-19 and healthy controls further supported this conclusion [36]. Despite similar viral loads, patients with mild and moderate COVID-19 expressed higher levels of anti-viral ISGs in nasal epithelia than patients with severe COVID-19 and healthy controls [36]. Macrophages in the nasopharyngeal swabs that more highly expressed inflammatory cytokine genes, e.g., *IL1β*, *TNF* and *CXCL8*, also characterized persons with severe COVID-19 [36]. The importance of the early induction of type 1 IFNs in the URT is further supported by an observation that persons with autoantibodies to type 1 IFN in nasal mucosa and blood are more likely to develop severe COVID-19 [37].

### 2.3. Antibodies in URT Immune Responses

Progress has also been made in determining the mechanisms of protection conferred by different classes of mucosal antibodies against SARS-CoV-2 in the URT. IgA and IgM antibodies in the nasal fluid and saliva were found to correlate with virus neutralization ability, and IgG antibodies in the nasal fluid with promoting phagocytosis in in vitro assays [38,39].

### 2.4. Implications for Early Protective URT Immunity

Overall, the findings reviewed in this section suggest that greater resistance to SARS-CoV-2 infection is achieved through better early innate and adaptive immune responses that more rapidly eliminate the virus in the URT. They also highlight an important role for IgA, IgG and IgM antibodies in the URT. The reported presence of greater numbers of sub-epithelial lymphoid follicles in the nasal mucosa of children [40] may also contribute to better immunity to SARS-CoV-2 infection, and this merits further investigation. SARS-CoV-2 is capable of interfering with innate immune responses (discussed further in Section 3), so that a preactivated innate immune system in the nasal mucosa may be better able to combat SARS-CoV-2 infections in children. An important corollary to these findings is that poorer nasal air conditioning in older adults [14,15] can exacerbate an already weaker URT immunity and make them more prone to SARS-CoV-2 infection and severe COVID-19 than children.

## 3. SARS-CoV-2 Mechanisms for Evading URT Immunity

SARS-CoV-2 interferes with type 1 and 3 IFN production in nasal epithelial cells, as described in Section 2 [28]. The SARS-CoV-2 protein NSP1 selectively promoted the degradation of cellular mRNA and inhibited nuclear mRNA export [41], while another protein ORF6 disrupted nuclear–cytoplasmic transport in both directions [42]. ORF6 also inhibited the MHC class 1 antigen presenting pathway by subverting the nuclear import of the trans-activator NLRC5 produced from type 2 IFNγ signaling—a process that has a critical function in adaptive immune responses [43]. Roles have also been ascribed to many other SARS-CoV-2 proteins in evading innate immunity in the URT and LRT [44]. Immune evasion by SARS-CoV-2 emphasizes the importance of a rapid and effective early immune response in the URT, produced, if necessary, by vaccination, to prevent infection and severe COVID-19.

An important feature of SARS-CoV-2 is that the virus evolves new strains with mutations that influence virus antigenicity and transmissibility. S from the Wuhan 2020 isolate, termed WAI/2020, has been the basis for many widely used vaccines. The SARS-CoV-2 strain B.1.351, or the beta variant, which rapidly became dominant initially in South Africa, developed many amino acid mutations that led to the partial evasion of COVID-19 and vaccine-induced neutralizing antibodies. One of the first experiments to test vaccine protection against a SARS-CoV-2 variant in a non-human primate model investigated B.1.351 using the Ad26.COV2.S vaccine developed by Johnson and Johnson [45]. The Ad26.COV2.S, an adenovirus-based vaccine, incorporates S from the WAI/2020 strain. It was used to immunize macaques intramuscularly followed by challenge with either the WAI/2020 strain or the B.1.351 strain of SARS-CoV-2. Although WAI/2020 S immunization led to significantly lower total serum and neutralizing antibody levels against B.1.351, the CD4+ T_H_ and CD8+ T_C_ cell responses to B.1.351 as well as the other common variants B.1.1.7 and P.1, when measured using IFNγ ELISPOT assays, were not significantly different. Importantly, challenge of the WAI/2020 S immunized macaques with B.1.351 virus resulted in robust cross-protection as characterized by the inhibition of viral replication in nasal swabs and bronchiolar lavage fluid, as well as lung histopathology. However, higher viral loads were initially observed upon challenge with B.1.351 in nasal swabs and bronchiolar lavage fluids [45], which is consistent with antigenic changes in S reducing early immunity.

The Omicron variant of SARS-CoV-2 has recently spread rapidly throughout the world, causing great concern. This variant has 32 mutations in S, of which 15 are in the RBD, and it is more rapidly transmissible than all other known variant strains [46]. In sera from convalescent patients infected with an early strain of SARS-CoV-2, virus-neutralizing antibody titers against Omicron were reduced by the greatest reported amount compared with many other SARS-CoV-2 variants of concern [46]. However, several very recent reports show that both CD4+ T_H_ and CD8+ T_C_ cell responses are adequately maintained against the Omicron variant after vaccination and recovery from COVID-19 [47,48,49,50,51]. Following vaccination with the early-strain S protein in BNT162b2, mRNA-1273, Ad26.CoV2.S or NVX- CoV2373 (a recombinant full-length S protein subunit vaccine) vaccines, 84% of CD4+ T_H_ and 85% of CD8+ T_C_ cell responses were preserved against the Omicron variant S protein [47]. Emerging data suggest that the Omicron variant causes clinically milder COVID-19 [52,53,54], which may partly reflect a strong contribution from T-cell-mediated protective immune responses after vaccination or prior COVID-19 infection.

## 4. Human Genetic Factors Influencing SARS-CoV-2 Infectivity in the URT

Whole-exome analysis recently identified a variant TMPRSS2 associated with less severe COVID-19 that had reduced S-cleaving activity [55]. The possibility that this variant TMPRSS2 may influence the course of SARS-CoV-2 infection in the URT requires further investigation. Genome-wide associations identifying large chromosomal regions that reduced or enhanced susceptibility to severe COVID-19 were previously highlighted [15]. Subsequent data suggest that the enhanced expression of a gene in one of the susceptibility regions, 3p21.31, codes for a leucine zipper transcription factor like 1 (LZTFL1) protein that promotes epithelial–mesenchymal transition in the pulmonary epithelium and more severe COVID-19 [56]. LZTFL1 is also expressed in the URT epithelium [57], and therefore, the possibility that this change may additionally enhance SARS-CoV-2 infection of the URT merits investigation.

However, a recent large international collaborative effort identified changes in a different gene in the 3p21.31 region, *SLC6A20*, that codes for an amino acid transporter associated with ACE2, and which appeared to enhance susceptibility to SARS-CoV-2 infection rather than the severity of COVID-19 [58]. Evidence from the same study also implicated ABO blood group genes highlighted previously [15,59,60], where one suggested explanation was that IgM antibodies to blood group A and B antigens in the URT mucosa of a blood group O host may bind to the A and B antigens on the virus envelope derived from the previous host and neutralize the virus. This same large study [58] also newly identified changes in a gene *PPP1R15A*, coding for a protein phosphatase regulatory subunit, as being responsible for susceptibility to infection rather than COVID-19 severity. The same study also confirmed roles for changes in other genes for proteins involved in the type 1 IFN induction of anti-viral immunity, which had been independently associated with susceptibility to developing severe COVID-19, e.g., the genes for the IFNα receptor [61,62] and oligoadenylate synthase [62] were associated with either symptomatic COVID-19 or both susceptibility to infection and symptomatic COVID-19 [58]. The susceptibility to initial infection of the URT cannot be easily differentiated from susceptibility to develop severe COVID-19 because both are linked [15], but genomic analysis based on the more detailed classification of disease phenotypes is proving to be helpful in this regard [63].

## 5. URT Immunity after SARS-CoV-2 Infection

Infection with SARS-CoV-2 can produce a wide spectrum of clinical symptoms, and an estimated 20–40% of global infections are asymptomatic, i.e., they show no overt clinical symptoms [64]. Asymptomatic infections are likely to be infections that are rapidly eliminated in the URT. The proportion of persons that remain asymptomatic after infection varies with age, health and immune status [64], and probably genetic background [15]. Asymptomatically infected persons can, however, transmit the virus, albeit over a shorter time period than symptomatic patients. Knowledge of the immune responses responsible for preventing the development of symptoms is important for elucidating the immune correlates of protection against COVID-19, the development of vaccines and advancing the understanding of URT immunity. All the effective early innate and adaptive immune responses in the URT discussed in Section 2 can be expected to contribute to the development of asymptomatic infections.

Recent findings in unvaccinated health care workers (HCWs) who remained persistently seronegative and uninfected in nasal swab PCR tests, but with likely repeated exposure to SARS-CoV-2, as shown by their T cells responding to S and other structural proteins, are particularly illuminating in this respect [65]. Mucosal antibodies in the URT of seronegative HCWs were not measured in this study. However, the seronegative HCWs had high levels of T cells with a memory phenotype, particularly CD4+ T_H_ cells, which recognized epitopes in NSP12 and NSP7 proteins of the viral replication–transcription complex as well as high serum levels of the IFNα-inducible protein IFI27, a protein that induces apoptosis in virus-infected cells and a likely marker for SARS-CoV-2 infection. The characterized SARS-CoV-2 CD4+ T_H_ cell epitopes had similar amino acid sequences to predicted CD4+ T_H_ cell epitopes in homologous proteins from common cold coronaviruses and demonstrable in vitro T_H_ cell cross-recognition [65]. These findings suggest that memory T cells primed with epitopes on more benign coronavirus replication-transcription complex proteins, and responding to cross-reactive SARS-CoV-2 epitopes, have an important role in rapidly terminating SARS-CoV-2 infections in the URT before the onset of noticeable clinical symptoms. It is pertinent in this context that proteins of the replication–transcription complex of SARS-CoV-2, which are encoded in ORF1 of its RNA genome, are among the earliest virus proteins synthesized during infection.

The preservation of cross-reactive CD4+ T_H_ and CD8+ T_C_ cell epitopes between SARS-CoV-2 variants, at least for S proteins [47,48,49,50,51] and early data from clinical protection against the Omicron variant [52,53,54], also suggests that previous infection or exposure to SARS-CoV-2 provides protection against developing more severe COVID-19 during a subsequent infection. COVID-19 or vaccination with S leads to the generation of S-epitope-specific memory CD4+ T_H_ cells and circulating follicular T_H_ cells that rapidly expand upon re-exposure to the S antigen [66]. Although the localization of these cells to the URT was not investigated, it seems likely that they function to enhance a memory-adaptive immune response in lymphoid tissue associated with the URT. Resident memory CD8+ T_C_ cells specific to influenza virus are found in the nasal mucosa after a resolved infection, and these efficiently control a second infection with the influenza virus [20,67]. Resident memory CD8+ T_C_ cells specific for SARS-CoV-2 persist in the nasal mucosa for at least two months after recovery from COVID-19 [68], and it seems likely that, analogous to influenza, they help resolve the infection in the URT and provide protective immunity in subsequent exposure to the virus.

Mucosal antibody responses to the SARS-CoV-2 proteins in the URT have been less well studied than serum antibody responses in COVID-19. Dimeric IgA, IgM and IgG are the major isotypes of antibodies in the URT mucosa. The mucosal levels of IgA and IgG antibodies to S did not always correlate with the levels of the corresponding antibody classes in blood during COVID-19 [69]. A recent detailed study of patients with mild COVID-19 and their household contacts measured IgA, IgG and IgM antibody levels in nasal mucosal fluid and serum to S, RBD and the nucleocapsid proteins, as well as viral loads in the URT [70]. The results showed that the viral load inversely correlated with antibody levels to S and RBD (with the strongest negative correlation seen for IgM anti-S antibodies) and that mucosal antibodies detectable on day three remained elevated for at least nine months after infection [70]. Observations that nasal IgA antibody levels correlate with SARS-CoV-2 neutralization in vitro and with less severe COVID-19 [36,38], and that neutralizing IgA antibodies in the URT are detectable at least 73d after infection [39], additionally illustrate the importance of adaptive immunity in the URT for protection against infection. The persistence of antibodies after SARS-CoV-2 infection is likely to be due to long-lived plasma cells that may reside in the bone marrow [71], and subsequent vaccination can further improve the neutralizing capability of such long-lived serum antibodies [72].

## 6. URT Immunity after Intramuscular COVID-19 Vaccination

### 6.1. Antibody Responses

Many COVID-19 vaccines in widespread use at present are based on mRNA, recombinant adenoviruses, recombinant proteins and inactivated virus vaccine platforms, and are delivered via intramuscular injection. Details of COVID-19 vaccines and their known immunological mechanisms of protection have been recently reviewed [73]. All currently used vaccines elicit serum antibodies able to react with S and RBD and neutralize SARS-CoV-2 infectivity in in vitro assays [73]. Findings from clinical trials of the ChAdOx1 nCov-19 intramuscularly injected vaccine developed by the University of Oxford and Astra-Zeneca and the Moderna mRNA1273 vaccine showed that antibody levels to RBD and S as well as virus neutralization titers in vaccinees correlated with protection from symptomatic infections with SARS-CoV-2, but correlations with asymptomatic infections could not be observed within the limits of the trials [74,75]. Observations on antibodies in the URT were not reported in these two trials [74,75].

More recently, it has been demonstrated that both the Pfizer/BioNTech BNT162b2 and Moderna mRNA1273 mRNA vaccines elicit URT antibodies in saliva [76]. Salivary IgG anti-S antibodies were found in all vaccine recipients, and IgA anti-S antibodies were found in a large proportion of vaccinees, after two doses of the vaccines [76]. More detailed results were obtained in the experimental vaccination of macaques [77]. Vaccination with the Moderna mRNA vaccine elicited IgG and IgA anti-S antibodies in nasal wash and bronchiolar lavage, with the nasal wash antibodies affording protection against virus replication [77]. The formation of IgA and IgG anti-S1 antibodies with neutralizing ability in the nasal cavity of vaccinees was reported after the intramuscular delivery of a different mRNA vaccine (Comirnaty) [78], but this was not observed with an intramuscularly administered, inactivated, whole-virus vaccine (CoronaVac) [78]. In a ferret model, the ChAdOx nCoV-19 adenovirus-vectored S vaccine delivered intramuscularly significantly reduced virus loads in nasal and oral washes of the animals when they were subsequently challenged intranasally with SARS-CoV-2 [79]. Dimeric IgA antibodies made in vitro against S are more efficient than monomeric IgA antibodies to S in virus neutralization assays in vitro [80]. Dimeric IgA is the usual secreted form of IgA in all mucosal surfaces.

The diminished protection afforded by vaccines based on early strain S against subsequent SARS-CoV-2 variants expressing multiple mutations in S and its RBD correlates with the ability of serum antibodies to neutralize virus infectivity in in vitro assays [45,46]. However, studies in non-human primate models with an adenovirus-vectored S protein vaccine (Ad26.COV2.S ) showed that vaccine efficacy is also likely to depend on anti-viral T cells [45].

### 6.2. T Cell Responses

Detailed results on T cell responses in persons vaccinated with the ChAdOx1 nCov-19 intramuscular vaccine were recently reported [81]. S-specific CD4+ T cell helper type 1 (T_H_1) and CD8+ T_C_ cell responses were increased in AZD1222-vaccinated adults of all ages after two doses of AZD1222, with many cells producing the cytokines IFNγ, TNF and IL2. The breadth of epitopes and the depth of recognition of particular epitopes were increased in the vaccinees. T cell responses in the URT of vaccinees were not investigated in these studies. It is pertinent, however, that SARS-CoV-2-specific memory CD8+ T_C_ cells with a follicular homing and tissue resident phenotype are found in the tonsils of unexposed children and adults, and these probably arose through the cross-recognition of other pathogens [82].

### 6.3. Summary

These findings are compatible with early anamnestic URT immune responses reducing SARS-CoV-2 infectivity in the URT of recipients of intramuscularly administered vaccines, which in turn lowers the incidence of symptomatic COVID-19. It is therefore desirable that trials of all COVID-19 vaccines administered intramuscularly are designed to examine immune responses in the URT as well as blood. This will help to better assess the elicited immune mechanisms from the point of view of protection being conferred at the earliest stage of infection by SARS-CoV-2.

## 7. URT Immunity after Intranasal COVID-19 Vaccination

A nasally administered COVID-19 vaccine will have the advantage of being delivered to the initial site of SARS-CoV-2 infection to generate locally protective immune responses. Effective URT immunity can then reduce virus multiplication and eliminate the virus as rapidly as possible at the primary infection site in the URT, thereby minimizing the infection of the LRT and severe disease. Intranasally administered live attenuated influenza vaccines, which have a long history of use in many countries [83], are a useful model for prospective intranasal COVID-19 vaccines.

Several adenovirus-vectored COVID-19 vaccines delivered intranasally have shown potential efficacy in animal models. A simian adenovirus 36 expressing S, termed ChAd-SARS-CoV-2-S, when given intranasally in a single dose to mice engineered to express human ACE2, elicited near-sterilizing immunity in the URT and LRT with systemic and mucosal immune responses. Virus-neutralizing IgA antibodies were present in bronchiolar lavage fluid, but nasal samples were not tested in this study [84]. Similarly, a single intranasal vaccination of mice with an adenovirus expressing RBD elicited systemic and mucosal immune responses, including mucosal IgA in bronchoalveolar fluid. Transgenic mice expressing human ACE2 were fully protected for at least six months from a lethal challenge with SARS-CoV-2 following this vaccination [85].

The efficacy and use of the ChAdOx1 nCov-19 intramuscularly injected vaccine developed by the University of Oxford and Astra-Zeneca has been widely reported [86,87,88]. Studies in ferret models showed that either intramuscular or intranasal immunization with this vaccine resulted in significantly reduced viral loads in nasal and oral fluids upon challenge with live virus [79]. Furthermore, the intranasal immunization of hamsters and macaques with ChAdOx1 nCov-19 followed by challenge with SARS-CoV-2 showed decreased viral loads in nasal wash and bronchiolar lavage fluids in macaques as well as the absence of detectable virus in lungs and reduced nasal virus loads in hamsters [89].

A recent promising approach described results from the use of a helper-dependent adenovirus vector producing secreted, soluble RBD (HD-Ad RBD) for the intranasal immunization of mice [90]. The HD-Ad RBD vaccine in a prime boost regimen elicited strong mucosal antibody responses in the LRT (IgG and IgA anti-RBD antibodies in bronchiolar lavage fluid), systemic immune responses (neutralizing serum IgG and IgA antibodies, IFNγ producing CD4+ T_H_ cells) and almost completely protected against lung inflammation in transgenic mice expressing human ACE2 on challenge with SARS-CoV-2 [90]. Importantly, the vaccinated transgenic mice expressing human ACE2 almost completely lacked a replicating virus in oropharyngeal swabs after challenge with SARS-CoV-2, also indicating effective protection in the URT. The HD-Ad RBD vaccine only produced transient inflammation upon immunization [90]. It has the advantage of being able to accommodate large gene inserts in the vector, so that multiple RBDs from SARS-CoV-2 variants of concern can be expressed.

In a related approach, the S1 protein subunit with appropriate adjuvants, used in the intramuscular priming and intranasal boosting immunization of macaques, showed no viral replication in the URT or LRT on SARS-CoV-2 challenge [91].

The widely used Pfizer/BioNTech BNT162b2 and Moderna mRNA1273 mRNA vaccines as well as other mRNA vaccines, administered intramuscularly, elicited potentially protective URT antibodies as described in Section 6 [76,77], but the formulation of mRNA vaccines for intranasal delivery is difficult and has yet to be described.

A different approach for eliciting mucosal immunity uses food-grade lactic acid bacteria as vectors. The oral immunization of rabbits and mice with *Lactococcus lactis* and *Lactobacillus* species expressing covalently or non-covalently bound pathogen proteins elicited systemic and gut mucosal immune responses [92,93]. Combined intranasal and oral immunization with *L. lactis* was also effective in eliciting mucosal IgA antibodies in the intestinal tract [94]. Recently, *Lactobacillus plantarum* expressing SARS-CoV-2 RBD on its surface was used for the nasal immunization of mice to show the production of IgA anti-RBD antibodies in nasal washing, bronchiolar lavage fluid and feces [95]. The disadvantages of using bacterial vectors for nasal immunization are that a dose of vaccine has to administered over several consecutive days for consistent responses, there is a need for a mature and non-senescent immune system for best responses [94] and there is possibly greater potential to elicit type 1 hypersensitivity reactions.

Other efforts are underway to develop intranasally delivered vaccines for COVID-19, but all of these await the completion of human trials and regulatory approval [96,97]. The possible development of safe, effective and inexpensive vaccines capable of large-scale manufacture for intranasal immunization, with the ability to accommodate antigens from multiple variants of SARS-CoV-2, can be crucially important for controlling the COVID-19 pandemic.

## 8. Conclusions

Early and effective innate and adaptive immune responses in the URT, generated either through vaccination or prior infection with SARS2-CoV-19, are important for controlling and eliminating virus replication in the URT, and thereby preventing severe COVID-19 due to infection of the LRT. Several established and potential protective immune mechanisms against SARS-CoV-2 infection in the URT are summarized in Table 1. Efforts to further advance knowledge on URT immunity in SARS-CoV-2 infection and COVID-19 vaccination are needed. SARS-CoV-2 variants that are more transmissible and better able to subvert URT immunity will continue to evolve in the community. COVID-19, like influenza, may therefore require the regular vaccination of vulnerable populations with appropriate variant antigens. Safe, effective and readily manufactured intranasally delivered vaccines can be particularly helpful in this respect.

## Figures and Tables

**Figure 1 viruses-14-00933-f001:**
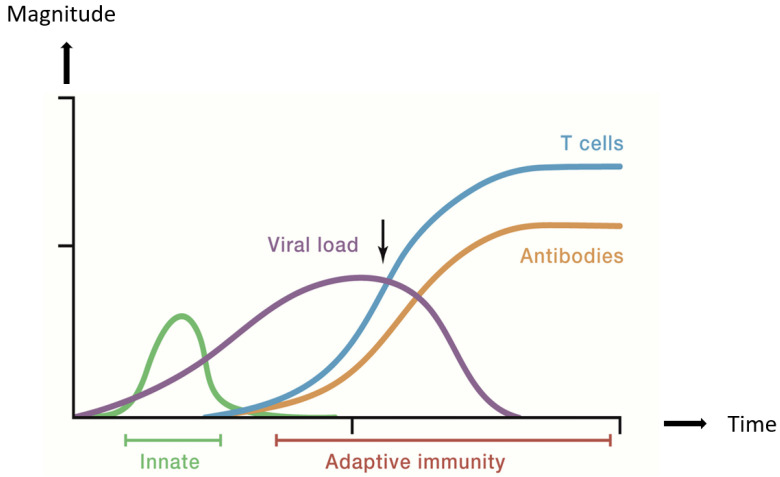
Generalized URT Immune Response and Viral Load in Mild COVID-19. Modified from Reference [16] with permission from Elsevier.

**Figure 2 viruses-14-00933-f002:**
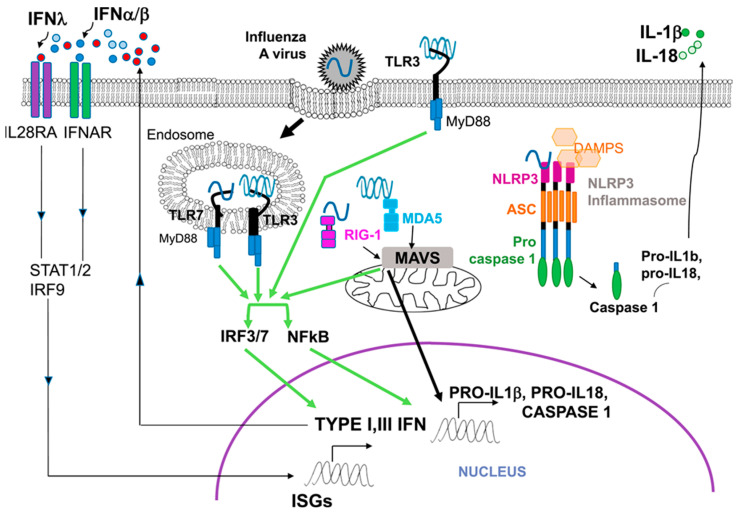
Principal Innate Immunity Signaling Pathways in Influenza A Virus Infection Viral RNA (a PAMP) in the endosome is sensed by the PRRs TLR3 and TLR7, and on the cell surface by TLR3. Viral RNA in the cytoplasm is recognized by the PRRs RIG-I and MDA-5, which then interact with the mitochondrial protein MAVS. These interactions between PAMPs and PRRs activate the transcription factors NF-κB and the interferon regulatory factors IRF3 and IRF7 to promote transcription of genes for the synthesis of type 1 (α, β), type 3 (λ) IFNs, and pro-inflammatory proteins. The IFNs bind to the infected epithelial cell’s own IFN receptors or IFN receptors on nearby cells to cause the synthesis of proteins from many interferon-stimulated genes (ISGs) that inhibit virus function in several ways. In parallel, cytosolic viral RNA and DAMPs interact with NLRP3 of the inflammasome complex to activate caspase-1 and produce the inflammatory cytokines IL-1β and IL-18. Reproduced as permitted under the creative commons license from Reference [19].

**Figure 3 viruses-14-00933-f003:**
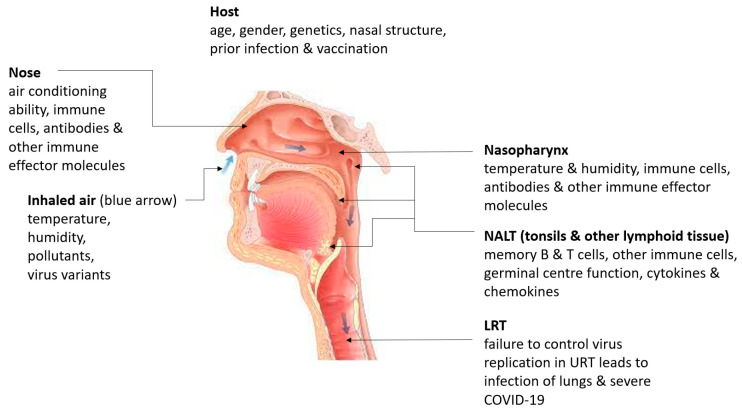
Overview of factors influencing immunity to SARS -CoV-2 in the upper respiratory tract. NALT—nasopharynx-associated lymphoid tissue. LRT- lower respiratory tract. Modified from Reference [15] under the creative commons license.

**Table 1 viruses-14-00933-t001:** Potential and established protective innate and adaptive immune mechanisms against SARS-CoV-2 in the upper respiratory tract.

Induction	Effector Cell or Molecule and Mechanism	Reference
1. **Innate immune responses** (References [13,15,18,19,27,98] for general detail)		
Virion entering the URT	Naturally occurring mucins, defensins and collectins that bind virions and prevent their binding and entry in epithelial cells	[99]
Altered surface of the virion and virus-infected cells	Complement activation through the alternate or lectin pathway to promote lysis and opsonization, inflammation	[99]
Pathogen-associated molecular pattern (PAMP) recognition by host cell pattern recognition receptors (PRRs)	(i) Production of type 1 (α, β) and type 3 (λ) interferons to induce an anti-viral state in infected and neighboring cells through inhibition of protein synthesis and mRNA degradation. Activation of phagocytic cells and dendritic cells (ii) Activation of inflammasome in macrophages and dendritic cells to produce IL-1, IL-6 and TNF that promote inflammatory responses in tissue, fever and the synthesis of acute phase proteins (iii) Macrophage and dendritic cell synthesis of IL-12 and IL-18 that activate NK cells to lyse virus-infected cells and enhance adaptive immune responses	[28,29,30,31,32,33,34,35,36,37,58,59,60,61,62,63]
Damage-associated molecular patterns (DAMPs) and PAMPs in infected cells	Activation of unconventional T cells (γδT, iNKT and MAIT) that in turn activate NK cells, phagocytes, dendritic cells and the adaptive immune response	[100]
2. **Adaptive immune responses** (References [15,16,73,98] for general detail)		
Secreted IgA antibodies in mucus	Preventing virion binding to epithelial cells through agglutination and neutralization of virions	[38,39,70,74,77,80]
IgG and IgM antibodies in mucus including IgM antibodies to A and B blood group antigens	Preventing virion binding to host cells through agglutination and neutralization, activation of complement through the classical pathway, promoting opsonization and phagocytosis, assisting NK cell killing through Fcγ receptors	[38,39,58,59,60,70,71,72,73,74,75,76,77,80]
CD4+ T_H_ lymphocytes, including memory cells recognizing SARS-CoV-2 epitopes and cross-reactive epitopes from related coronaviruses	Activating B cells, promoting immunoglobulin class switching and affinity maturation, secretion of cytokines such as IFNγ that activate phagocytes and NK cells and upregulating major histocompatibility complex molecules and antigen presentation. Additionally, direct cytotoxicity on infected cells	[65,66]
CD8+ T_C_ lymphocytes, including memory cells recognizing SARS-CoV-2 epitopes and cross-reactive epitopes from related coronaviruses	Cytotoxicity to virus-infected cells mediated through granzyme, perforin, etc.	[31,67,68,82]

## Data Availability

All data supporting the conclusions of this article are included within the article.

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
