# Peer review of "Innate and Adaptive Immune Responses in the Upper Respiratory Tract and the Infectivity of SARS-CoV-2"

_viruses, 2022, doi:10.3390/v14050933_

Round 1

Reviewer 1 Report

Though this review important to the field, many articles relating to Innate and Adaptive Immune Responses to COVID19. Several additions and changes are to be made including figures, illustrations and more detailed description of T cells, B cells and NK cells  and other mechanisms of innate and adaptive immune responses are all required.

Reviewer 2 Report

The authors aimed to review literature about innate and adaptive immune responses in the upper respiratory tract and the infectivity of SARS-CoV-2.

The study covers some issues that have been overlooked in other similar topics. The structure of the manuscript appears adequate and well divided in the sections. Moreover, the study is easy to follow, but few issues should be improved. Some of the comments that would improve the overall quality of the study are:

1-) The manuscript needs grammar correction. Please also check typos thorough the text;

2-) Conclusion Section: This paragraph required a general revision to eliminate redundant sentences and to add some "take-home message".

Reviewer 3 Report

It is quite interesting article, reviewing the mechanisms of immune response in respiratory infections and COVID-19. 

The conclusions of this review are obvious and therefore not so substantial in the issue of confronting the COVID-19 pandemic. 

I suggest changing the figure 1 to colored one and more clear. 

Reviewer 4 Report

Major revisions in a few sections:

  1. Sections 1, 2 and 3 of the review are largely based on a previous review published by the same author (Reference no. 14) and has redundant content (Please find the paper attached for your reference). The tables and figures are reproduced from said reference and have no original content. This section can either be removed or shortened appropriately by citing the references

  1. Section 4 summarizes findings from studies in primary cell culture models and from studies delineating immune responses to SARS CoV2 in children and adults. The ideas in the section appear to be disjointed and can be summarized separately.

3. Sections 5-9 appropriately summarize recent knowledge regarding immune response to SARS CoV2 infection and various vaccines against COVID-19. 

  1. Conclusion section adequately summarizes information in above sections while providing future directions.

Round 2

Reviewer 1 Report

Comments are addressed. Some english need to be improved and typos including the scientific terms.